# Minimum Weight Perfect Matching
# via Blossom Belief Propagation

**Sungsoo Ahn**[*]      **Sejun Park**[*]      **Michael Chertkov**[†]      **Jinwoo Shin**[*]
[*]School of Electrical Engineering,
Korea Advanced Institute of Science and Technology, Daejeon, Korea
[†]Theoretical Division and Center for Nonlinear Studies,
Los Alamos National Laboratory, Los Alamos, USA
[*]{sungsoo.ahn, sejun.park, jinwoos}@kaist.ac.kr    [†]chertkov@lanl.gov

## Abstract

Max-product Belief Propagation (BP) is a popular message-passing algorithm for computing a Maximum-A-Posteriori (MAP) assignment over a distribution represented by a Graphical Model (GM). It has been shown that BP can solve a number of combinatorial optimization problems including minimum weight matching, shortest path, network flow and vertex cover under the following common assumption: the respective Linear Programming (LP) relaxation is tight, i.e., no integrality gap is present. However, when LP shows an integrality gap, no model has been known which can be solved systematically via sequential applications of BP. In this paper, we develop the first such algorithm, coined Blossom-BP, for solving the minimum weight matching problem over arbitrary graphs. Each step of the sequential algorithm requires applying BP over a modified graph constructed by contractions and expansions of blossoms, i.e., odd sets of vertices. Our scheme guarantees termination in $O(n^2)$ of BP runs, where $n$ is the number of vertices in the original graph. In essence, the Blossom-BP offers a distributed version of the celebrated Edmonds' Blossom algorithm by jumping at once over many sub-steps with a single BP. Moreover, our result provides an interpretation of the Edmonds' algorithm as a sequence of LPs.

## 1   Introduction

Graphical Models (GMs) provide a useful representation for reasoning in a number of scientific disciplines [1, 2, 3, 4]. Such models use a graph structure to encode the joint probability distribution, where vertices correspond to random variables and edges specify conditional dependencies. An important inference task in many applications involving GMs is to find the most-likely assignment to the variables in a GM, i.e., Maximum-A-Posteriori (MAP). Belief Propagation (BP) is a popular algorithm for approximately solving the MAP inference problem and it is an iterative, message passing one that is exact on tree structured GMs. BP often shows remarkably strong heuristic performance beyond trees, i.e., over loopy GMs. Furthermore, BP is of a particular relevance to large-scale problems due to its potential for parallelization [5] and its ease of programming within the modern programming models for parallel computing, e.g., GraphLab [6], GraphChi [7] and OpenMP [8].

The convergence and correctness of BP was recently established for a certain class of loopy GM formulations of several classical combinatorial optimization problems, including matching [9, 10, 11], perfect matching [12], shortest path [13], independent set [14], network flow [15] and vertex cover [16]. The important common feature of these models is that BP converges to a correct assignment when the Linear Programming (LP) relaxation of the combinatorial optimization is tight, i.e., when it shows no integrality gap. The LP tightness is an inevitable condition to guarantee the performance of BP and no combinatorial optimization instance has been known where BP would be used to solve

problems without the LP tightness. On the other hand, in the LP literature, it has been extensively studied how to enforce the LP tightness via solving multiple intermediate LPs that are systematically designed, e.g., via the cutting-plane method [21]. Motivated by these studies, we pose a similar question for BP, "how to enforce correctness of BP, possibly by solving multiple intermediate BPs". In this paper, we show how to resolve this question for the minimum weight (or cost) perfect matching problem over arbitrary graphs.

**Contribution.** We develop an algorithm, coined Blossom-BP, for solving the minimum weight matching problem over an arbitrary graph. Our algorithm solves multiple intermediate BPs until the final BP outputs the solution. The algorithm is sequential, where each step includes running BP over a 'contracted' graph derived from the original graph by contractions and infrequent expansions of blossoms, i.e., odd sets of vertices. To build such a scheme, we first design an algorithm, coined Blossom-LP, solving multiple intermediate LPs. Second, we show that each LP is solvable by BP using the recent framework [16] that establishes a generic connection between BP and LP. For the first part, cutting-plane methods solving multiple intermediate LPs for the minimum weight matching problem have been discussed by several authors over the past decades [17, 18, 19, 20] and a provably polynomial-time scheme was recently suggested [21]. However, LPs in [21] were quite complex to solve by BP. To address the issue, we design much simpler intermediate LPs that allow utilizing the framework of [16].

We prove that Blossom-BP and Blossom-LP guarantee to terminate in $O(n^2)$ of BP and LP runs, respectively, where $n$ is the number of vertices in the graph. To establish the polynomial complexity, we show that intermediate outputs of Blossom-BP and Blossom-LP are equivalent to those of a variation of the Blossom-V algorithm [22] which is the latest implementation of the Blossom algorithm due to Kolmogorov. The main difference is that Blossom-V updates parameters by maintaining disjoint tree graphs, while Blossom-BP and Blossom-LP implicitly achieve this by maintaining disjoint cycles, claws and tree graphs. Notice, however, that these combinatorial structures are auxiliary, as required for proofs, and they do not appear explicitly in the algorithm descriptions. Therefore, they are much easier to implement than Blossom-V that maintains complex data structures, e.g., priority queues. To the best of our knowledge, Blossom-BP and Blossom-LP are the simplest possible algorithms available for solving the problem in polynomial time. Our proof implies that in essence, Blossom-BP offers a distributed version of the Edmonds' Blossom algorithm [23] jumping at once over many sub-steps of Blossom-V with a single BP.

The subject of solving convex optimizations (other than LP) via BP was discussed in the literature [24, 25, 26]. However, we are not aware of any similar attempts to solve Integer Programming, via sequential application of BP. We believe that the approach developed in this paper is of a broader interest, as it promises to advance the challenge of designing BP-based MAP solvers for a broader class of GMs. Furthermore, Blossom-LP stands alone as providing an interpretation for the Edmonds' algorithm in terms of a sequence of tractable LPs. The Edmonds' original LP formulation contains exponentially many constraints, thus naturally suggesting to seek for a sequence of LPs, each with a subset of constraints, gradually reducing the integrality gap to zero in a polynomial number of steps. However, it remained illusive for decades: even when the bipartite LP relaxation of the problem has an integral optimal solution, the standard Edmonds' algorithm keeps contracting and expanding a sequence of blossoms. As we mentioned earlier, we resolve the challenge by showing that Blossom-LP is (implicitly) equivalent to a variant of the Edmonds' algorithm with three major modifications: (a) parameter-update via maintaining cycles, claws and trees, (b) addition of small random corrections to weights, and (c) initialization using the bipartite LP relaxation.

**Organization.** In Section 2, we provide backgrounds on the minimum weight perfect matching problem and the BP algorithm. Section 3 describes our main result – Blossom-LP and Blossom-BP algorithms, where the proof is given in Section 4.

## 2 Preliminaries

### 2.1 Minimum weight perfect matching

Given an (undirected) graph $G = (V, E)$, a matching of $G$ is a set of vertex-disjoint edges, where a perfect matching additionally requires to cover every vertices of $G$. Given integer edge weights (or costs) $w = [w_e] \in \mathbb{Z}^{|E|}$, the minimum weight (or cost) perfect matching problem consists in computing a perfect matching which minimizes the summation of its associated edge weights. The

problem is formulated as the following IP (Integer Programming):

$$\text{minimize} \quad w \cdot x \quad \text{subject to} \quad \sum_{e \in \delta(v)} x_e = 1, \quad \forall v \in V, \qquad x = [x_e] \in \{0,1\}^{|E|} \quad (1)$$

Without loss of generality, one can assume that weights are strictly positive.[1] Furthermore, we assume that IP (1) is feasible, i.e., there exists at least one perfect matching in $G$. One can naturally relax the above integer constraints to $x = [x_e] \in [0,1]^{|E|}$ to obtain an LP (Linear Programming), which is called the bipartite relaxation. The integrality of the bipartite LP relaxation is not guaranteed, however it can be enforced by adding the so-called blossom inequalities [22]:

$$\text{minimize} \quad w \cdot x$$

$$\text{subject to} \quad \sum_{e \in \delta(v)} x_e = 1, \quad \forall v \in V, \qquad \sum_{e \in \delta(S)} x_e \geq 1, \quad \forall S \in \mathcal{L}, \qquad x = [x_e] \in [0,1]^{|E|},$$

$$(2)$$

where $\mathcal{L} \subset 2^V$ is a collection of odd cycles in $G$, called blossoms, and $\delta(S)$ is a set of edges between $S$ and $V \setminus S$. It is known that if $\mathcal{L}$ is the collection of all the odd cycles in $G$, then LP (2) always has an integral solution. However, notice that the number of odd cycles is exponential in $|V|$, thus solving LP (2) is computationally intractable. To overcome this complication we are looking for a tractable subset of $\mathcal{L}$ of a polynomial size which guarantees the integrality. Our algorithm, searching for such a tractable subset of $\mathcal{L}$ is iterative: at each iteration it adds or subtracts a blossom.

## 2.2 Belief propagation for linear programming

A joint distribution of $n$ (binary) random variables $Z = [Z_i] \in \{0,1\}^n$ is called a Graphical Model (GM) if it factorizes as follows: for $z = [z_i] \in \Omega^n$,

$$\Pr[Z = z] \propto \prod_{i \in \{1,\dots,n\}} \psi_i(z_i) \prod_{\alpha \in F} \psi_\alpha(z_\alpha),$$

where $\{\psi_i, \psi_\alpha\}$ are (given) non-negative functions, the so-called factors; $F$ is a collection of subsets

$$F = \{\alpha_1, \alpha_2, \dots, \alpha_k\} \subset 2^{\{1,2,\dots,n\}}$$

(each $\alpha_j$ is a subset of $\{1, 2, \dots, n\}$ with $|\alpha_j| \geq 2$); $z_\alpha$ is the projection of $z$ onto dimensions included in $\alpha$.[2] In particular, $\psi_i$ is called a variable factor. Assignment $z^*$ is called a maximum-a-posteriori (MAP) solution if $z^* = \arg\max_{z \in \{0,1\}^n} \Pr[z]$. Computing a MAP solution is typically computationally intractable (i.e., NP-hard) unless the induced bipartite graph of factors $F$ and variables $z$, so-called factor graph, has a bounded treewidth. The max-product Belief Propagation (BP) algorithm is a popular simple heuristic for approximating the MAP solution in a GM, where it iterates messages over a factor graph. BP computes a MAP solution exactly after a sufficient number of iterations, if the factor graph is a tree and the MAP solution is unique. However, if the graph contains loops, BP is not guaranteed to converge to a MAP solution in general. Due to the space limitation, we provide detailed backgrounds on BP in the supplemental material.

Consider the following GM: for $x = [x_i] \in \{0,1\}^n$ and $w = [w_i] \in \mathbb{R}^n$,

$$\Pr[X = x] \propto \prod_i e^{-w_i x_i} \prod_{\alpha \in F} \psi_\alpha(x_\alpha), \tag{3}$$

where $F$ is the set of non-variable factors and the factor function $\psi_\alpha$ for $\alpha \in F$ is defined as

$$\psi_\alpha(x_\alpha) = \begin{cases} 1 & \text{if } A_\alpha x_\alpha \geq b_\alpha, \ C_\alpha x_\alpha = d_\alpha \\ 0 & \text{otherwise} \end{cases},$$

for some matrices $A_\alpha, C_\alpha$ and vectors $b_\alpha, d_\alpha$. Now we consider Linear Programming (LP) corresponding to this GM:

$$\begin{aligned} \text{minimize} \quad & w \cdot x \\ \text{subject to} \quad & \psi_\alpha(x_\alpha) = 1, \quad \forall \alpha \in F, \qquad x = [x_i] \in [0,1]^n. \end{aligned} \tag{4}$$

One observes that the MAP solution for GM (3) corresponds to the (optimal) solution of LP (4) if the LP has an integral solution $x^* \in \{0,1\}^n$. Furthermore, the following sufficient conditions relating max-product BP to LP are known [16]:

**Theorem 1** *The max-product BP applied to GM (3) converges to the solution of LP (4) if the following conditions hold:*

    *C1. LP (4) has a unique integral solution $x^* \in \{0,1\}^n$, i.e., it is tight.*

    *C2. For every $i \in \{1, 2, \ldots, n\}$, the number of factors associated with $x_i$ is at most two, i.e., $|F_i| \leq 2$.*

    *C3. For every factor $\psi_\alpha$, every $x_\alpha \in \{0,1\}^{|\alpha|}$ with $\psi_\alpha(x_\alpha) = 1$, and every $i \in \alpha$ with $x_i \neq x_i^*$, there exists $\gamma \subset \alpha$ such that*

$$|\{j \in \{i\} \cup \gamma : |F_j| = 2\}| \leq 2$$

$$\psi_\alpha(x'_\alpha) = 1, \qquad \text{where } x'_k = \begin{cases} x_k & \text{if } k \notin \{i\} \cup \gamma \\ x_k^* & \text{otherwise} \end{cases}.$$

$$\psi_\alpha(x''_\alpha) = 1, \qquad \text{where } x''_k = \begin{cases} x_k & \text{if } k \in \{i\} \cup \gamma \\ x_k^* & \text{otherwise} \end{cases}.$$

## 3 Main result: Blossom belief propagation

In this section, we introduce our main result – an iterative algorithm, coined Blossom-BP, for solving the minimum weight perfect matching problem over an arbitrary graph, where the algorithm uses the max-product BP as a subroutine. We first describe the algorithm using LP instead of BP in Section 3.1, where we call it Blossom-LP. Its BP implementation is explained in Section 3.2.

### 3.1 Blossom-LP algorithm

Let us modify the edge weights: $w_e \leftarrow w_e + n_e$, where $n_e$ is an i.i.d. random number chosen in the interval $\left[0, \frac{1}{|V|}\right]$. Note that the solution of the minimum weight perfect matching problem (1) remains the same after this modification because the overall noise does not exceed 1. The Blossom-LP algorithm updates the following parameters iteratively.

    ◦ $\mathcal{L} \subset 2^V$: a *laminar* collection of odd cycles in $G$.

    ◦ $y_v, y_S$: $v \in V$ and $S \in \mathcal{L}$.

In the above, $\mathcal{L}$ is called laminar if for every $S, T \in \mathcal{L}$, $S \cap T = \emptyset$, $S \subset T$ or $T \subset S$. We call $S \in \mathcal{L}$ an *outer* blossom if there exists no $T \in \mathcal{L}$ such that $S \subset T$. Initially, $\mathcal{L} = \emptyset$ and $y_v = 0$ for all $v \in V$. The algorithm iterates between Step **A** and Step **B** and terminates at Step **C**.

**Blossom-LP algorithm**

---

**A. Solving LP on a contracted graph.** First construct an auxiliary (contracted) graph $G^\dagger = (V^\dagger, E^\dagger)$ by contracting every outer blossom in $\mathcal{L}$ to a single vertex, where the weights $w^\dagger = [w_e^\dagger : e \in E^\dagger]$ are defined as

$$w_e^\dagger = w_e - \sum_{v \in V : v \notin V^\dagger, e \in \delta(v)} y_v - \sum_{S \in \mathcal{L} : v(S) \notin V^\dagger, e \in \delta(S)} y_S, \qquad \forall\, e \in E^\dagger.$$

We let $v(S)$ be the blossom vertex in $G^\dagger$ coined as the contracted graph and solve the following LP:

$$
\begin{aligned}
\text{minimize} \quad & w^\dagger \cdot x \\
\text{subject to} \quad & \sum_{e \in \delta(v)} x_e = 1, \quad \forall\, v \in V^\dagger, \ v \text{ is a non-blossom vertex} \\
& \sum_{e \in \delta(v)} x_e \geq 1, \quad \forall\, v \in V^\dagger, \ v \text{ is a blossom vertex} \\
& x = [x_e] \in [0,1]^{|E^\dagger|}.
\end{aligned}
\tag{5}
$$

**B. Updating parameters.** After we obtain a solution $x = [x_e : e \in E^\dagger]$ of LP (5), the parameters are updated as follows:

(a) If $x$ is integral, i.e., $x \in \{0,1\}^{|E^\dagger|}$ and $\sum_{e \in \delta(v)} x_e = 1$ for all $v \in V^\dagger$, then proceed to the termination step **C**.

(b) Else if there exists a blossom $S$ such that $\sum_{e \in \delta(v(S))} x_e > 1$, then we choose one of such blossoms and update
$$\mathcal{L} \leftarrow \mathcal{L} \backslash \{S\} \qquad \text{and} \qquad y_v \leftarrow 0, \quad \forall\, v \in S.$$
Call this step 'blossom $S$ expansion'.

(c) Else if there exists an odd cycle $C$ in $G^\dagger$ such that $x_e = 1/2$ for every edge $e$ in it, we choose one of them and update
$$\mathcal{L} \leftarrow \mathcal{L} \cup \{V(C)\} \qquad \text{and} \qquad y_v \leftarrow \frac{1}{2} \sum_{e \in E(C)} (-1)^{d(e,v)} w_e^\dagger, \quad \forall v \in V(C),$$
where $V(C), E(C)$ are the set of vertices and edges of $C$, respectively, and $d(v,e)$ is the graph distance from vertex $v$ to edge $e$ in the odd cycle $C$. The algorithm also remembers the odd cycle $C = C(S)$ corresponding to every blossom $S \in \mathcal{L}$.

If (b) or (c) occur, go to Step **A**.

**C. Termination.** The algorithm iteratively expands blossoms in $\mathcal{L}$ to obtain the minimum weighted perfect matching $M^*$ as follows:

(i) Let $M^*$ be the set of edges in the original $G$ such that its corresponding edge $e$ in the contracted graph $G^\dagger$ has $x_e = 1$, where $x = [x_e]$ is the (last) solution of LP (5).

(ii) If $\mathcal{L} = \emptyset$, output $M^*$.

(iii) Otherwise, choose an outer blossom $S \in \mathcal{L}$, then update $G^\dagger$ by expanding $S$, i.e. $\mathcal{L} \leftarrow \mathcal{L} \backslash \{S\}$.

(iv) Let $v$ be the vertex in $S$ covered by $M^*$ and $M_S$ be a matching covering $S \backslash \{v\}$ using the edges of odd cycle $C(S)$.

(v) Update $M^* \leftarrow M^* \cup M_S$ and go to Step (ii).

---

An example of the evolution of $\mathcal{L}$ is described in the supplementary material. We provide the following running time guarantee for this algorithm, which is proven in Section 4.

**Theorem 2** *Blossom-LP outputs the minimum weight perfect matching in $O(|V|^2)$ iterations.*

### 3.2 Blossom-BP algorithm

In this section, we show that the algorithm can be implemented using BP. The result is derived in two steps, where the first one consists in the following theorem proven in the supplementary material due to the space limitation.

**Theorem 3** *LP (5) always has a half-integral solution $x^* \in \left\{0, \frac{1}{2}, 1\right\}^{|E^\dagger|}$ such that the collection of its half-integral edges forms disjoint odd cycles.*

Next let us design BP for obtaining the half-integral solution of LP (5). First, we duplicate each edge $e \in E^\dagger$ into $e_1, e_2$ and define a new graph $G^\ddagger = (V^\dagger, E^\ddagger)$ where $E^\ddagger = \{e_1, e_2 : e \in E^\ddagger\}$. Then, we build the following equivalent LP:

$$
\begin{aligned}
\text{minimize} \quad & w^\ddagger \cdot x \\
\text{subject to} \quad & \sum_{e \in \delta(v)} x_e = 2, \quad \forall\, v \in V^\dagger, \ v \text{ is a non-blossom vertex} \\
& \sum_{e \in \delta(v)} x_e \geq 2, \quad \forall\, v \in V^\dagger, \ v \text{ is a blossom vertex} \\
& x = [x_e] \in [0,1]^{|E^\dagger|},
\end{aligned}
\tag{6}
$$

where $w_{e_1}^{\ddagger} = w_{e_2}^{\ddagger} = w_e^{\dagger}$. One can easily observe that solving LP (6) is equivalent to solving LP (5) due to our construction of $G^{\ddagger}, w^{\ddagger}$, and LP (6) always have an integral solution due to Theorem 3. Now, construct the following GM for LP (6):

$$\Pr[X = x] \ \propto \ \prod_{e \in E^{\ddagger}} e^{w_e^{\ddagger} x_e} \prod_{v \in V^{\dagger}} \psi_v(x_{\delta(v)}), \tag{7}$$

where the factor function $\psi_v$ is defined as

$$\psi_v(x_{\delta(v)}) = \begin{cases} 1 & \text{if } v \text{ is a non-blossom vertex and } \sum_{e \in \delta(v)} x_e = 2 \\ 1 & \text{else if } v \text{ is a blossom vertex and } \sum_{e \in \delta(v)} x_e \geq 2 \\ 0 & \text{otherwise} \end{cases}.$$

For this GM, we derive the following corollary of Theorem 1 proven in the supplementary material due to the space limitation.

**Corollary 4** *If LP* (6) *has a unique solution, then the max-product BP applied to GM* (7) *converges to it.*

The uniqueness condition stated in the corollary above is easy to guarantee by adding small random noises to edge weights. Corollary 4 shows that BP can compute the half-integral solution of LP (5).

## 4 Proof of Theorem 2

First, it is relatively easy to prove the correctness of Blossom-BP, as stated in the following lemma.

**Lemma 5** *If Blossom-LP terminates, it outputs the minimum weight perfect matching.*

**Proof.** We let $x^{\dagger} = [x_e^{\dagger}], y^{\ddagger} = [y_v^{\ddagger}, y_S^{\ddagger} : v \notin V^{\dagger}, v(S) \notin V^{\dagger}]$ denote the parameter values at the termination of Blossom-BP. Then, the strong duality theorem and the complementary slackness condition imply that

$$x_e^{\dagger}(w^{\dagger} - y_u^{\dagger} - y_v^{\dagger}) = 0, \quad \forall e = (u, v) \in E^{\dagger}. \tag{8}$$

where $y^{\dagger}$ be a dual solution of $x^{\dagger}$. Here, observe that $y^{\dagger}$ and $y^{\ddagger}$ cover $y$-variables inside and outside of $V^{\dagger}$, respectively. Hence, one can naturally define $y^* = [y_v^{\dagger} \ y_u^{\ddagger}]$ to cover all $y$-variables, i.e., $y_v, y_S$ for all $v \in V, S \in \mathcal{L}$. If we define $x^*$ for the output matching $M^*$ of Blossom-LP as $x_e^* = 1$ if $e \in M^*$ and $x_e^* = 0$ otherwise, then $x^*$ and $y^*$ satisfy the following complementary slackness condition:

$$x_e^* \left( w_e - y_u^* - y_v^* - \sum_{S \in \mathcal{L}} y_S^* \right) = 0, \quad \forall e = (u, v) \in E, \qquad y_S^* \left( \sum_{e \in \delta(S)} x_e^* - 1 \right) = 0, \quad \forall S \in \mathcal{L},$$

where $\mathcal{L}$ is the last set of blossoms at the termination of Blossom-BP. In the above, the first equality is from (8) and the definition of $w^{\dagger}$, and the second equality is because the construction of $M^*$ in Blossom-BP is designed to enforce $\sum_{e \in \delta(S)} x_e^* = 1$. This proves that $x^*$ is the optimal solution of LP (2) and $M^*$ is the minimum weight perfect matching, thus completing the proof of Lemma 5. $\square$

To guarantee the termination of Blossom-LP in polynomial time, we use the following notions.

**Definition 1** *Claw is a subset of edges such that every edge in it shares a common vertex, called center, with all other edges, i.e., the claw forms a star graph.*

**Definition 2** *Given a graph $G = (V, E)$, a set of odd cycles $\mathcal{O} \subset 2^E$, a set of claws $\mathcal{W} \subset 2^E$ and a matching $M \subset E$, $(\mathcal{O}, \mathcal{W}, M)$ is called cycle-claw-matching decomposition of $G$ if all sets in $\mathcal{O} \cup \mathcal{W} \cup \{M\}$ are disjoint and each vertex $v \in V$ is covered by exactly one set among them.*

To analyze the running time of Blossom-BP, we construct an iterative auxiliary algorithm that outputs the minimum weight perfect matching in a bounded number of iterations. The auxiliary algorithm outputs a cycle-claw-matching decomposition at each iteration, and it terminates when the cycle-claw-matching decomposition corresponds to a perfect matching. We will prove later that the auxiliary algorithm and Blossom-LP are equivalent and, therefore, conclude that the iteration of Blossom-LP is also bounded.

To design the auxiliary algorithm, we consider the following dual of LP (5):

$$\text{minimize} \quad \sum_{v \in V^\dagger} y_v$$

$$\text{subject to} \quad w_e^\dagger - y_v - y_u \geq 0, \quad \forall e = (u,v) \in E^\dagger, \quad y_{v(S)} \geq 0, \quad \forall S \in \mathcal{L}. \tag{9}$$

Next we introduce an auxiliary iterative algorithm which updates iteratively the blossom set $\mathcal{L}$ and also the set of variables $y_v, y_S$ for $v \in V, S \in \mathcal{L}$. We call edge $e = (u,v)$ 'tight' if $w_e - y_u - y_v - \sum_{S \in \mathcal{L}: e \in \delta(S)} y_S = 0$. Now, we are ready to describe the auxiliary algorithm having the following parameters.

○ $G^\dagger = (V^\dagger, E^\dagger)$, $\mathcal{L} \subset 2^V$, and $y_v, y_S$ for $v \in V, S \in \mathcal{L}$.

○ $(\mathcal{O}, \mathcal{W}, M)$: A cycle-claw-matching decomposition of $G^\dagger$

○ $T \subset G^\dagger$: A tree graph consisting of $+$ and $-$ vertices.

Initially, set $G^\dagger = G$ and $\mathcal{L}, T = \emptyset$. In addition, set $y_v, y_S$ by an optimal solution of LP (9) with $w^\dagger = w$ and $(\mathcal{O}, \mathcal{W}, M)$ by the cycle-claw-matching decomposition of $G^\dagger$ consisting of tight edges with respect to $[y_v, y_S]$. The parameters are updated iteratively as follows.

**The auxiliary algorithm**

---

**Iterate the following steps until $M$ becomes a perfect matching:**

1. Choose a vertex $r \in V^\dagger$ from the following rule.

   **Expansion.** If $\mathcal{W} \neq \emptyset$, choose a claw $W \in \mathcal{W}$ of center blossom vertex $c$ and choose a non-center vertex $r$ in $W$. Remove the blossom $S(c)$ corresponding to $c$ from $\mathcal{L}$ and update $G^\dagger$ by expanding it. Find a matching $M'$ covering all vertices in $W$ and $S(c)$ except for $r$ and update $M \leftarrow M \cup M'$.

   **Contraction.** Otherwise, choose a cycle $C \in \mathcal{O}$, add and remove it from $\mathcal{L}$ and $\mathcal{O}$, respectively. In addition, $G^\dagger$ is also updated by contracting $C$ and choose the contracted vertex $r$ in $G^\dagger$ and set $y_r = 0$.

   Set tree graph $T$ having $r$ as $+$ vertex and no edge.

2. Continuously increase $y_v$ of every $+$ vertex $v$ in $T$ and decrease $y_v$ of $-$ vertex $v$ in $T$ by the same amount until one of the following events occur:

   **Grow.** If a tight edge $(u,v)$ exists where $u$ is a $+$ vertex of $T$ and $v$ is covered by $M$, find a tight edge $(v,w) \in M$. Add edges $(u,v), (v,w)$ to $T$ and remove $(v,w)$ from $M$ where $v, w$ becomes $-, +$ vertices of $T$, respectively.

   **Matching.** If a tight edge $(u,v)$ exists where $u$ is a $+$ vertex of $T$ and $v$ is covered by $C \in \mathcal{O}$, find a matching $M'$ that covers $T \cup C$. Update $M \leftarrow M \cup M'$ and remove $C$ from $\mathcal{O}$.

   **Cycle.** If a tight edge $(u,v)$ exists where $u, v$ are $+$ vertices of $T$, find a cycle $C$ and a matching $M'$ that covers $T$. Update $M \leftarrow M \cup M'$ and add $C$ to $\mathcal{O}$.

   **Claw.** If a blossom vertex $v(S)$ with $y_{v(S)} = 0$ exists, find a claw $W$ (of center $v(S)$) and a matching $M'$ covering $T$. Update $M \leftarrow M \cup M'$ and add $W$ to $\mathcal{W}$.

   If **Grow** occurs, resume the step 2. Otherwise, go to the step 1.

---

Note that the auxiliary algorithm updates parameters in such a way that the number of vertices in every claw in the cycle-claw-matching decomposition is 3 since every $-$ vertex has degree 2. Hence, there exists a unique matching $M'$ in the expansion step. Furthermore, the existence of a cycle-claw-matching decomposition at the initialization can be guaranteed using the complementary slackness condition and the half-integrality of LP (5). We establish the following lemma for the running time of the auxiliary algorithm, where its proof is given in the supplemental material due to the space limitation.

**Lemma 6** *The auxiliary algorithm terminates in $O(|V|^2)$ iterations.*

Now we are ready to prove the equivalence between the auxiliary algorithm and the Blossom-LP, i.e., prove that the numbers of iterations of Blossom-LP and the auxiliary algorithm are equal. To this end, given a cycle-claw-matching decomposition $(\mathcal{O}, \mathcal{W}, M)$, observe that one can choose the corresponding $x = [x_e] \in \{0, 1/2, 1\}^{|E^\dagger|}$ that satisfies constraints of LP (5):

$$x_e = \begin{cases} 1 & \text{if } e \text{ is an edge in } \mathcal{W} \text{ or } M \\ \frac{1}{2} & \text{if } e \text{ is an edge in } \mathcal{O} \\ 0 & \text{otherwise} \end{cases}.$$

Similarly, given a half-integral $x = [x_e] \in \{0, 1/2, 1\}^{|E^\dagger|}$ that satisfies constraints of LP (5), one can find the corresponding cycle-claw-matching decomposition. Furthermore, one can also define weight $w^\dagger$ in $G^\dagger$ for the auxiliary algorithm as Blossom-LP does:

$$w_e^\dagger = w_e - \sum_{v \in V : v \notin V^\dagger, e \in \delta(v)} y_v - \sum_{S \in \mathcal{L} : v(S) \notin V^\dagger, e \in \delta(S)} y_S, \qquad \forall\, e \in E^\dagger. \tag{10}$$

In the auxiliary algorithm, $e = (u, v) \in E^\dagger$ is tight if and only if $w_e^\dagger - y_u^\dagger - y_v^\dagger = 0$. Under these equivalences in parameters between Blossom-LP and the auxiliary algorithm, we will use the induction to show that cycle-claw-matching decompositions maintained by both algorithms are equal at every iteration, as stated in the following lemma whose proof is given in the supplemental material due to the space limitation..

**Lemma 7** *Define the following notation:*

$$y^\dagger = [y_v : v \in V^\dagger] \qquad \text{and} \qquad y^\ddagger = [y_v, y_S : v \in V, v \notin V^\dagger, S \in \mathcal{L}, v(S) \notin V^\dagger],$$

*i.e., $y^\dagger$ and $y^\ddagger$ are parts of $y$ which involves and does not involve in $V^\dagger$, respectively. Then, the Blossom-LP and the auxiliary algorithm update parameters $\mathcal{L}, y^\ddagger$ equivalently and output the same cycle-claw-decomposition of $G^\dagger$ at each iteration.*

The above lemma implies that Blossom-LP also terminates in $O(|V|^2)$ iterations due to Lemma 6. This completes the proof of Theorem 2. The equivalence between the half-integral solution of LP (5) in Blossom-LP and the cycle-claw-matching decomposition in the auxiliary algorithm implies that LP (5) is always has a half-integral solution, and hence, one of Steps **B.**(a), **B.**(b) or **B.**(c) always occurs.

## 5   Conclusion

The BP algorithm has been popular for approximating inference solutions arising in graphical models, where its distributed implementation, associated ease of programming and strong parallelization potential are the main reasons for its growing popularity. This paper aims for designing a polynomial-time BP-based scheme solving the maximum weigh perfect matching problem. We believe that our approach is of a broader interest to advance the challenge of designing BP-based MAP solvers in more general GMs as well as distributed (and parallel) solvers for large-scale IPs.

**Acknowledgement.**   This work was supported by Institute for Information & communications Technology Promotion(IITP) grant funded by the Korea government(MSIP) (No.R0132-15-1005), Content visual browsing technology in the online and offline environments. The work at LANL was carried out under the auspices of the National Nuclear Security Administration of the U.S. Department of Energy under Contract No. DE-AC52-06NA25396.

## Footnotes

[1]If some edges have negative weights, one can add the same positive constant to all edge weights, and this does not alter the solution of IP (1).

[2]For example, if $z = [0,1,0]$ and $\alpha = \{1,3\}$, then $z_\alpha = [0,0]$.

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
