[Supplementary Material]

# Supplementary Material: Minimum Weight Perfect Matching via Blossom Belief Propagation

## A  Background on Max-Product Belief Propagation

The max-product Belief Propagation (BP) algorithm is a popular heuristic for approximating the MAP assignment in a GM. BP is implemented iteratively; at each iteration $t$, it maintains four messages

$$\{m_{\alpha \to i}^t(c), m_{i \to \alpha}^t(c) : c \in \{0,1\}\}$$

between every variable $z_i$ and every associated $\alpha \in F_i$, where $F_i := \{\alpha \in F : i \in \alpha\}$; that is, $F_i$ is a subset of $F$ such that all $\alpha$ in $F_i$ include the $i^{th}$ position of $z$ for any given $z$. The messages are updated as follows:

$$m_{\alpha \to i}^{t+1}(c) \;=\; \max_{z_\alpha : z_i = c} \psi_\alpha(z_\alpha) \prod_{j \in \alpha \setminus i} m_{j \to \alpha}^t(z_j) \tag{11}$$

$$m_{i \to \alpha}^{t+1}(c) \;=\; \psi_i(c) \prod_{\alpha' \in F_i \setminus \alpha} m_{\alpha' \to i}^t(c). \tag{12}$$

where each $z_i$ only sends messages to $F_i$; that is, $z_i$ sends messages to $\alpha_j$ only if $\alpha_j$ selects/includes $i$. The outer-term in the message computation (11) is maximized over all possible $z_\alpha \in \{0,1\}^{|\alpha|}$ with $z_i = c$. The inner-term is a product that only depends on the variables $z_j$ (excluding $z_i$) that are connected to $\alpha$. The message-update (12) from variable $z_i$ to factor $\psi_\alpha$ is a product containing all messages received by $\psi_\alpha$ in the previous iteration, except for the message sent by $z_i$ itself.

Given a set of messages $\{m_{i \to \alpha}(c), m_{\alpha \to i}(c) : c \in \{0,1\}\}$, the so-called BP max-marginals are computed as follows:

$$b_i[z_i] \;=\; \psi_i(z_i) \prod_{\alpha \in F_i} m_{\alpha \to i}(z_i). \tag{13}$$

This BP algorithm outputs $z^{BP} = [z_i^{BP}]$ where

$$z_i^{BP} = \begin{cases} 1 & \text{if } b_i[1] > b_i[0] \\ ? & \text{if } b_i[1] = b_i[0] \\ 0 & \text{if } b_i[1] < b_i[0] \end{cases}.$$

It is known that $z^{BP}$ converges to a MAP assignment after a sufficient number of iterations, if the factor graph is a tree and the MAP assignment is unique. However, if the graph contains loops, the BP algorithm is not guaranteed to converge to a MAP assignment in general.

## B  Proof of Theorem 3

For the proof of Theorem 3, once we show the half-integrality of LP (5), it is easy to check that the half-integral edges forms disjoint odd cycles. Hence, it suffices to show that every vertex of the polytope consisting of constraints of LP (5) is always half-integral. To this end, we show the following lemma.

**Lemma 8** *Let $A = [A_{ij}] \in \{0,1\}^{m \times m}$ be an invertible 0-1 matrix whose row has at most two non-zero entires. Then, each entry $A_{ij}^{-1}$ of $A^{-1}$ is in $\left\{0, \pm 1, \pm \frac{1}{2}\right\}$.*

**Proof.** Suppose there exists a row in $A$ with one non-zero entry. Then, one can assume that it is the first row of $A$ and $A_{11} = 1$ without loss of generality. Hence, $A_{11}^{-1} = 1$, $A_{1i}^{-1} = 0$ for $i \neq 1$ and the first column of $A^{-1}$ has only $0$ and $\pm 1$ entries since each row of $A$ has at most two non-zero entries. This means that one can proceed the proof of Lemma 8 for the submatrix of $A$ deleting the first row and column. Therefore, one can assume that each row of $A$ contains exactly two non-zero entries.

We construct a graph $G = (V, E)$ such that

$$V = [m] := \{1, 2, \ldots, m\} \qquad \text{and} \qquad E = \{(j, k) : a_{ij} = a_{ik} = 1 \text{ for some } i \in V\},$$

i.e., each row $A_{i[m]} = (A_{i1}, \ldots, A_{im})$ and each column $A_{[m]i} = (A_{1i}, \ldots, A_{mi})^T$ correspond to an edge and a vertex of $G$, respectively. Since $A$ is invertible, one can notice that $G$ does not contain an even cycle as well as a path between two distinct odd cycles (including two odd cycles share a vertex). Therefore, each connected component of $G$ has at most one odd cycle. Consider the $i$-th column $A_{[m]i}^{-1} = (A_{1i}^{-1}, \ldots, A_{mi}^{-1})^T$ of $A^{-1}$ and we have

$$A_{i[m]} A_{[m]i}^{-1} = 1 \qquad \text{and} \qquad A_{j[m]} A_{[m]i}^{-1} = 0 \quad \text{for } j \neq i, \tag{14}$$

i.e., $A_{[m]i}^{-1}$ assigns some values on $V$ such that the sum of values on two end-vertices of the edge corresponding to the $k$-th row of $A$ is 1 and 0 if $k = i$ and $k \neq i$, respectively.

Let $e = (u, v) \in E$ be the edge corresponding to the $i$-th row of $A$.

- First, consider the case when $e$ is not in an odd cycle of $G$. Since each component of $G$ contains at most one odd cycle, one can assume that the component of $u$ is a tree in the graph $G \setminus e$. We will find the entries of $A^{-1}$ satisfying (14). Choose $A_{wi}^{-1} = 0$ for all vertex $w$ not in the component. and $A_{ui}^{-1} = 1$. Since the component forms a tree, one can set $A_{wi}^{-1} = 1$ or $-1$ for every vertex $w \neq u$ in the component to satisfy (14). This implies that $A_{[m]i}^{-1}$ consists of 0 and $\pm 1$.

- Second, consider the case when $e$ is in an odd cycle of $G$. We will again find the entries of $A^{-1}$ satisfying (14). Choose $A_{ui}^{-1} = A_{vi}^{-1} = \frac{1}{2}$ and $A_{wi}^{-1} = 0$ for every vertex $w$ not in the component containing $e$. Then, one can choose $A_{[m]i}^{-1}$ satisfying (14) by assigning $A_{wi}^{-1} = \frac{1}{2}$ or $-\frac{1}{2}$ for vertex $w \neq u, v$ in the component containing $e$. Therefore, $A_{[m]i}^{-1}$ consists of 0 and $\pm \frac{1}{2}$. $\square$

This completes the proof of Lemma 8.

Consider a vertex $x \in [0, 1]^{|E^\dagger|}$ of the polytope consisting of constraints of LP (5). Then, there exists a linear system of equalities such that $x$ is its unique solution where each equality is either $x_e = 0$, $x_e = 1$ or $\sum_{e \in \delta(v)} x_e = 1$. One can plug $x_e = 0$ and $x_e = 1$ into the linear system, reducing it to $Ax = b$ where $A$ is an invertible 0-1 matrix whose column contains at most two non-zero entries. Hence, from Lemma 8, $x$ is half-integral. This completes the proof of Theorem 3.

## C  Proof of Corollary 4

The proof of Corollary 4 will be completed using Theorem 1. If LP (6) has a unique solution, LP (6) has a unique and integral solution by Theorem 3, i.e., Condition *C1* of Theorem 1. LP (6) satisfies Condition *C2* as each edge is incident with two vertices. Now, we need to prove that LP (6) satisfies Condition *C3* of Theorem 1. Let $x^*$ be a unique optimal solution of LP (6). Suppose $v$ is a non-blossom vertex and $\psi_v(x_{\delta(v)}) = 1$ for some $x_{\delta(v)} \neq x_{\delta(v)}^*$. If $x_e \neq x_e^* = 1$ for $e \in \delta(v)$, there exist $f \in \delta(v)$ such that $x_f \neq x_f^* = 0$. Similarly, If $x_e \neq x_e^* = 0$ for $e \in \delta(v)$, there exists $f \in \delta(v)$ such that $x_f \neq x_f^* = 1$. Then, it follows that

$$\psi_v(x_{\delta}'(v)) = 1, \qquad \text{where } x_{e'}' = \begin{cases} x_{e'} & \text{if } e' \notin \{e, f\} \\ x_{e'}^* & \text{otherwise} \end{cases}.$$

$$\psi_v(x_{\delta}'(v)) = 1, \qquad \text{where } x_{e'}' = \begin{cases} x_{e'} & \text{if } e' \in \{e, f\} \\ x_{e'}^* & \text{otherwise} \end{cases}.$$

Suppose $v$ is a blossom vertex and $\psi_v(x_{\delta(v)}) = 1$ for some $x_{\delta(v)} \neq x^*_{\delta(v)}$. If $x_e \neq x^*_e = 1$ for $e \in \delta(v)$, choose $f \in \delta(v)$ such that $x_f \neq x^*_f = 0$ if it exists. Otherwise, choose $f = e$. Similarly, If $x_e \neq x^*_e = 0$ for $e \in \delta(v)$, choose $f \in \delta(v)$ such that $x_f \neq x^*_f = 1$ if it exists. Otherwise, choose $f = e$. Then, it follows that

$$\psi_v(x'_\delta(v)) = 1, \qquad \text{where } x'_{e'} = \begin{cases} x_{e'} & \text{if } e' \notin \{e, f\} \\ x^*_{e'} & \text{otherwise} \end{cases}.$$

$$\psi_v(x'_\delta(v)) = 1, \qquad \text{where } x'_{e'} = \begin{cases} x_{e'} & \text{if } e' \in \{e, f\} \\ x^*_{e'} & \text{otherwise} \end{cases}.$$

## D  Proof of Lemma 6

In this section, we prove that the number of iterations of the auxiliary algorithm is bounded by $O(|V|^2)$. To this end, let $(\mathcal{O}, \mathcal{W}, M)$ be the cycle-claw-matching decomposition of $G^\dagger$ and $N = |\mathcal{O}| + |\mathcal{W}|$ at some iteration of the algorithm. We first prove that $|\mathcal{O}| + |\mathcal{W}|$ does not increase at every iteration. At Step 1, the algorithm deletes an element in either $\mathcal{O}$ or $\mathcal{W}$ and hence, $|\mathcal{O}| + |\mathcal{W}| = N - 1$. On the other hand, at Step 2, one can observe that the algorithm run into one of the following scenarios with respect to $|\mathcal{O}| + |\mathcal{W}|$:

**Grow.** $|\mathcal{O}| + |\mathcal{W}| = N - 1$
**Matching.** $|\mathcal{O}| + |\mathcal{W}| = N - 2$
**Cycle.** $|\mathcal{O}| + |\mathcal{W}| = N$
**Claw.** $|\mathcal{O}| + |\mathcal{W}| = N$

Therefore, the total number of odd cycles and claws at Step 2 does not increase as well.

From now on, we define $\{t_1, t_2, \cdots : t_i \in \mathbb{Z}\}$ to be indexes of iterations when **Matching** occurs at Step 2, and we call the set of iterations $\{t : t_i \leq t < t_{i+1}\}$ as the $i$-th *stage*. We will show that the length of each stage is $O(|V|)$, i.e., for all $i$,

$$|t_i - t_{i+1}| = O(|V|). \tag{15}$$

This implies that the auxiliary algorithm terminates in $O(|V|^2)$ iterations since the total number of odd cycles and claws at the initialization is $O(|V|)$ and it decrease by two if **Matching** occurs. To this end, we prove the following key lemmas.

**Claim 9** *At every iteration of the auxiliary algorithm, there exist no path consisting of tight edges between two vertices $v_1, v_2 \in V^\dagger$ where each $v_i$ is either a blossom vertex $v(S)$ with $y_S = 0$ or a (blossom or non-blossom) vertex in an odd cycle consisted of tight edges.*

**Proof.** First observe that $w^\dagger$ (see (10) for its definition) is updated only at **Contraction** and **Expansion** of Step 1. If **Contraction** occurs, there exist a cycle $C$ to be contracted before Step 1. Then one can observe that before the contraction, for every vertex $v$ in $C$, $y_v$ is expressed as a linear combination of $w^\dagger$:

$$y_v = \frac{1}{2} \sum_{e \in E(C)} (-1)^{d_C(e,v)} w^\dagger_e, \tag{16}$$

where $d_C(v, e)$ is the graph distance from vertex $v$ to edge $e$ in the odd cycle $C$. Moreover $w^\dagger$ is updated after the contraction as

$$\begin{cases} w^\dagger_e \leftarrow w^\dagger_e - y_v & \text{if } v \text{ is in the cycle } C \text{ and } e \in \delta(v) \\ w^\dagger_e \leftarrow w^\dagger_e & \text{otherwise} \end{cases}.$$

Thus the updated value $w^\dagger_e$ can be expressed as a linear combination of the old values $w^\dagger$ where each coefficient is uniquely determined by $G^\dagger$. One can show the same conclusion similarly when **Expansion** occurs. Therefore one conclude the following.

♣ Each value $w^\dagger_e$ at any iteration can be expressed as a linear combination of the original weight values $w$ where each coefficient is uniquely determined by the prior history in $G^\dagger$.

To derive a contradiction, we assume there exist a path $P$ consisting of tight edges between two vertices $v_1$ and $v_2$ where each $v_i$ is either a blossom vertex $v(S)$ with $y_S = 0$ or a vertex in an odd cycle consisting of tight edges. Consider the case where $v_1$ and $v_2$ are in cycle $C_1$ and $C_2$ consisting of tight edges, where other cases can be argued similarly. Then one can observe that there exists a linear relationship between $y_v$ and $y_u$ and $w^\dagger$:

$$y_{v_1} + (-1)^{d_P(v_2, v_1)} y_{v_2} = \sum_{e \in P} (-1)^{d_P(e, v_1)} w_e^\dagger \tag{17}$$

where $d_P(v_2, v_1)$ and $d_P(e, v_1)$ is the graph distance from $v_1$ to $v_2$ and $e$, respectively, in the path $P$. Since $v_1, v_2$ are in cycles $C_1, C_2$, respectively, we can apply (16). From this observation, (17) and ♣, there exists a linear relationship among the original weight values $w$, where each coefficient is uniquely determined by the prior history in $G^\dagger$. This is impossible since the number of possible scenarios in the history of $G^\dagger$ is finite, whereas we add continuous random noises to $w$. This completes the proof of Claim 9. □

**Claim 10** *Consider a $+$ vertex $v \in V^\dagger$ at some iteration of the auxiliary algorithm. Then, at the first iteration afterward where $v$ becomes a $-$ vertex or is removed from $V^\dagger$ (i.e., due to the contraction of a blossom), it is connected to an odd cycle $C \in \mathcal{O}$ via an even-sized alternating path consisting of tight edges with respect to matching $M$ whenever each iteration starts during the same stage. Here, $\mathcal{O}$ and $M$ are from the cycle-claw-decomposition.*

**Proof.** To this end, suppose that a $+$ vertex $v$ at the $t^\dagger$-th iteration first becomes a $-$ vertex or is removed from $V^\dagger$ at the $t^\ddagger$-th iteration where $t^\dagger, t^\ddagger$-th iterations are in the same stage. First observe that if $v$ is removed from $G^\dagger$ at the $t^\ddagger$-th iteration, there exist a cycle in $\mathcal{O}$ that includes it at the start of the $t^\ddagger$-th iteration, resulting a zero-sized alternating path between such vertex and cycle, i.e., the conclusion of Lemma 10 holds. Now, for the other case, i.e., $v$ becomes a $-$ vertex at the $t^\ddagger$-th iteration, we will prove the following.

★ For any $t$-th iteration with $t^\dagger \le t < t^\ddagger$, one of the followings holds:

1. The vertex $v$ becomes a $+$ vertex during the $t$-th iteration. Moreover, $v$ either becomes a $+$ vertex during the $(t+1)$-th iteration or $v$ becomes connected to some cycle $C$ in $\mathcal{O}$ via an even-sized alternating path $P$ consisting of tight edges at the start of $(t+1)$-th iteration.

2. The vertex $v$ is not in the tree $T$ during the $t$-th iteration. Moreover, if $v$ is connected to some cycle $C$ in $\mathcal{O}$ via an even-sized alternating path $P$ consisting of tight edges at the start of $t$-th iteration, $v$ remains connected to cycle $C$ in $\mathcal{O}$ via an even-sized alternating path $P$ consisted of tight edges at the start of $(t + 1)$-th iteration, i.e. the algorithm parameters associated with $P$ and $C$ are not updated during the $t$-th iteration.

For ★ − 1, observe that if $v$ becomes a $+$ vertex during the $t$-th iteration, the iteration terminates with one of the following scenarios:

I. The iteration terminates with **Matching**. This contradicts to the assumption that $t^\dagger, t^\ddagger$-th iterations are in the same stage, i.e., no **Matching** occurs during the $t$-th iteration.

II. The iteration terminates with **Cycle**. The vertex $v$ is connected to the cycle newly added to $\mathcal{O}$ via an even-sized alternating path consisting of tight edges in tree $T$ at the start of the next (i.e., $(t + 1)$-th) iteration.

III. The iteration terminates with **Claw**. The vertex $v$ becomes a $+$ vertex of tree $T$ of the next (i.e., $(t + 1)$-th) iteration. This is due to the following reasons. After **Claw**, the algorithm expands the center vertex of newly made claw $W$ by **Expansion** in the next iteration. Then, there exists an even-sized alternating path $P_W$ from $r$ to $v$ consisted of tight edges in the newly constructed tree $T$. Furthermore, edges in $P_W$ are continuously added to $T$ by **Grow** without modifying parameter $y$ in Step 2 until $v$ becomes a $+$ vertex in $T$. This is because **Claw** and **Cycle** are impossible to occur due to Claim 9.

For ★ − 2, in order to derive a contradiction, assume that a vertex $v$ violates ★ − 2 at some iteration, i.e. the algorithm parameters associated to the even-sized alternating path $P$ and the cycle $C$ in the

statement of $\bigstar - 2$ are updated during the iteration. Observe that the algorithm parameters are updated due to one of the following scenarios:

I. The cycle $C$ is contracted. If $v$ is in $C$, $v$ no longer remains in $V^\dagger$ and contradicts to the assumption that $v$ remains in $V^\dagger$. If $v$ is not in $C$, $v$ becomes a $+$ vertex in tree $T$ after continuously adding edges of $P$ by **Grow** without modifying parameter $y$ due to Claim 9. This contradicts to the assumption of $\bigstar - 2$ that $v$ is not in tree $T$ during the $t$-th iteration.

II. A vertex in $C$ is added to tree $T$. Then, **Matching** occurs, i.e. the new stage starts. This contradicts to the assumption that $t^\dagger, t^\ddagger$-th iterations are in the same stage.

III. An edge in $P$ is added to tree $T$. Then, there exists a vertex $u$ in $P$ that first became a $-$ vertex among vertices in $P$, and it either (a) has an even-sized alternating path $P'$ to $C$ consisting of tight edges or (b) has an odd-sized alternating path $P'$ to $v$ consisting of tight edges. For (a), the edges in $P'$ are continuously added to $T$ without modifying parameter $y$ by Claim 9 and **Matching** occurs. This contradicts to the assumption again. For (b), $P'$ are added to $T$ without modifying parameter $y$ due Claim 9, and $v$ is added to tree $T$ as a $+$ vertex. This contradicts to the assumption of $\bigstar - 2$ that $v$ is not in tree $T$ during the $t$-th iteration.

Therefore, $\bigstar$ holds. One can observe that there exists $t^* \in (t^\dagger, t^\ddagger)$ such that at the $t^*$-th iteration, $v$ last becomes a $+$ vertex before the $t^\ddagger$-th iteration, i.e. $v$ is not in tree $T$ during $t$-th iteration for $t^* < t < t^\ddagger$. Then $v$ is connected to some cycle $C$ in $\mathcal{O}$ via an even length alternating path $P$ at $(t^*+1)$-th iteration and such path and cycle remains unchanged during $t$-th iteration for $t^* < t \leq t^\ddagger$ due to $\bigstar$. This completes the proof of Claim 10. $\qquad\square$

Now we aim for proving (15). To this end, we claim the following.

$\spadesuit$ A $+$ vertex of $V^\dagger$ at some iteration cannot be a $-$ one (whenever it appears in $V^\dagger$) afterward in the same stage.

For proving $\spadesuit$, we assume that a $+$ vertex $v \in V^\dagger$ at the $t$-th iteration violates $\spadesuit$ to derive a contradiction, i.e., it becomes a $-$ one in some tree $T$ during $t'$-th iteration in the same stage. Without loss of generality, one can assume that the vertex $v$ has the minimum value of $t' - t$ among such vertices violating $\spadesuit$. We consider two cases: (a) $v$ is always contained in $V^\dagger$ afterward in the same stage, and (b) $v$ is removed from $V^\dagger$ (at least once, due to the contraction of a blossom containing $v$) afterward in the same stage. First consider the case (a). Then, due to the assumption of the case (a) and Claim 10, there exist a path $P$ from $v$ to a cycle $C \in \mathcal{O}$ when the $t'$-th iteration starts. Then, one can observe that in order to add $v$ to tree $T$ as a $-$ vertex, it must be the first vertex in path $P$ added to $T$ by **Grow** during the $t$-iteration. Furthermore, tree $T$ keeps continuing to perform **Grow** afterward using tight edges of path $P$ without modifying parameter $y$ until **Matching** occurs, i.e., the new stage starts. This is because **Claw** and **Cycle** are impossible to occur before **Matching** due to Claim 9. Hence, it contradicts to the assumption that $t$ and $t'$ are in the same stage, and completes the proof of $\spadesuit$ for the case (a). Now we consider the case (b), i.e., $v$ is removed from $V^\dagger$ due to the contraction of a blossom $S \in \mathcal{L}$. In this case, the blossom vertex $v(S) \in V^\dagger$ must be expanded before $v$ becomes a $-$ vertex. However, $v(S)$ becomes a $+$ vertex after contracting $S$ and a $-$ vertex before expanding $v(S)$, i.e., $v(S)$ also violates $\spadesuit$. This contradicts to the assumption that the vertex $v$ has the minimum value of $t' - t$ among vertices violating $\spadesuit$, and completes the proof of $\spadesuit$. Due to $\spadesuit$, a blossom cannot expand after contraction in the same stage, where we remind that a blossom vertex becomes a $+$ one after contraction and a $-$ one before expansion. This implies that the number contractions and expansions in the same stage is $O(|V|)$, which leads to (15) and completes the proof of Lemma 6.

# E   Proof of Lemma 7

Initially, it is trivial. Now we assume the induction hypothesis that $\mathcal{L}, y^\ddagger$ and the cycle-claw-decomposition are equivalent between both algorithms at the previous iteration. First, it is easy to observe that $\mathcal{L}$ is updated equivalently since it is only decided by the cycle-claw-decomposition at the previous iteration in both algorithms. Next, it is also easy to check that $y^\ddagger$ is updated equivalently

since (a) if we remove a blossom $S$ from $\mathcal{L}$, it is trivial and (b) if we add a blossom $S = V(C)$ for some cycle $C$ to $\mathcal{L}$, $y^{\ddagger}$ is uniquely decided by $C$ and $w^{\dagger}$ in both algorithms.

In the remaining of this section, we will show that once $\mathcal{L}, y^{\ddagger}$ are updated equivalently, the cycle-claw-decomposition also changes equivalently in both algorithms. Observe that $G^{\dagger}, w^{\dagger}$ only depends on $\mathcal{L}, y^{\ddagger}$. In addition, $y^{\dagger}$ maintained by the auxiliary algorithm also satisfies constraints of LP (9). Consider the cycle-claw-matching decomposition $(\mathcal{O}, \mathcal{W}, M)$ of the auxiliary algorithm, and the corresponding $x = [x_e] \in \{0, 1/2, 1\}^{|E^{\dagger}|}$ that satisfies constraints of LP (5). Then, $x$ and $y^{\dagger}$ satisfy the complementary slackness condition:

$$x_e(w_e^{\dagger} - y_u^{\dagger} - y_v^{\dagger}) = 0, \qquad \forall e = (u, v) \in E^{\dagger}$$

$$y_{v(S)}^{\dagger}\left( \sum_{e \in \delta(v(S))} x_e - 1 \right) = 0, \qquad \forall S \in \mathcal{L},$$

where the first equality is because the cycle-claw-matching decomposition consists of tight edges and the second equality is because every claw maintained by the auxiliary algorithm has its center vertex $v(S)$ with $y_{v(S)} = 0$ for some $S \in \mathcal{L}$. Therefore, $x$ is an optimal solution of LP (5), i.e., the cycle-claw-decomposition is updated equivalently in both algorithms. This completes the proof of Lemma 7.

# F  Example of evolution of Blossoms under Blossom-LP

(a) Initial graph

(b) Solution of LP (5) in the 1st iteration

(c) Solution of LP (5) in the 2nd iteration

(d) Solution of LP (5) in the 3rd iteration

(e) Solution of LP (5) in the 4th iteration

(f) Solution of LP (5) in the 5th iteration

(g) Output matching

Figure 1: Example of evolution of Blossoms under Blossom-LP, where solid and dashed lines correspond to $1$ and $\frac{1}{2}$ solutions of LP (5), respectively.