[Reviews · NeurIPS 2015]

Submitted by Assigned_Reviewer_1

Overview: The manuscript introduces an approach to solving the minimum weight perfect matching problem via a sequence of LPs that can themselves be efficiently solved using belief propagation.

The established algorithm has many interesting connections and interpretations - one among them is that it "jumps" over many sub-steps of Edmonds' Blossom algorithm with a single run of belief propagation.

The construction of the algorithm is presented as follows:

1) At first, Blossom-LP is introduced, which solves the problem through a sequence of linear programs. (Similar algorithms, though with different LP formulations, are already known from the literature [22])

2) Then, it is shown how these intermediate LPs can be formulated as a graphical model for which BP is guaranteed to find an integral solution. (This is based on recent theoretical results of [16]).

The algorithm that solves each such LP via BP is called Blossom-BP.

3) Finally, an auxiliary algorithm is given that is more amenable to theoretical analysis.

It is first shown that this algorithm terminates correctly in O ( |V|^2 ) time, and then its equivalence to Blossom-LP (and hence Blossom-BP) is established.

Positive points: + The resulting algorithm seems interesting and practical, in particular since highly optimized implementations of BP are possible (in particular, it can easily be parallelized) + The theoretical analysis of the algorithm is very thorough; the established O ( |V|^2 ) complexity is a non-trivial result. + The manuscript contains plenty of non-trivial novel contributions, in addition to the innovative application of recent results of [16] and [22] + The manuscript is very well-written and reveals many interesting connections to related algorithms

Negative points: - The manuscript is theory-only.

At this point, it is hard to judge if an efficient implementation of Blossom-BP would be competitive with Blossom-V.

I am looking forward to first numerical experiments.
Summary: This is a very well-written, theory-only manuscript that introduces a novel approach to solving minimum weight perfect matching using iterated belief propagation.

The result is important, in that it provides an algorithm that establishes an optimal solution in O( |V|^2 ) belief propagation runs, and in that it provides an interpretation of Edmonds' algorithm as a sequence of LPs.

Submitted by Assigned_Reviewer_2

Adding some figures to illustrate the main contributions/algorithms is encouraged. For instance, the auxiliary algorithm may be easily demonstrated by a few figures.
Summary: The work is to propose an algorithm (Blossom-BP) to solve the minimum weight matching problem over arbitrary graphs. The theoretical foundation of the work is solid and well presented.

Submitted by Assigned_Reviewer_3

The authors describe an algorithm to solve the general MWM problem by incorporating blossom constraints (through a series of linear subproblems than can be solved by a BP subroutine).

I found that the paper was a bit tough to follow due to the page constraints (much of the content was in the supplemental material), and it could use some proof-reading to fix typos and improve the overall flow.

Despite this, the authors do present novel contributions.

It remains unclear whether similar strategies can be adapted to other settings.

General comments:

1. Is the overall complexity of the method mentioned in the paper?

How does it compare to Edmonds' Blossom algorithm?

2. The idea of sequentially adding constraints has been studied before (cycle constraints in particular). For example, "D. Sontag, D. K. Choe, Y. Li. Efficiently Searching for Frustrated Cycles in MAP Inference. Uncertainty in Artificial Intelligence (UAI) 28, Aug. 2012." While these methods use heuristics to pick cycles constraints to add, they have a similar flavor and may be worth citing.

3. For the half integrality of C-LP, I feel like I have seen this (or a very similar result before), but I could be wrong. The authors should double check the classical results about MWMs.

4. Many parts of the write-up could be improved.

For example, you discuss modifying the edges weights of the problem on page 4, but you don't explain why this is the case (until much later and you don't even refer back).

More generally, it's a bit difficult to carefully follow why the described approach actually solves the problem.

This is probably due to the space constraints but makes it tough to read nonetheless.

Summary: The authors describe an algorithm to solve the general MWM problem by incorporating blossom constraints (through a series of linear subproblems than can be solved by a BP subroutine).

I found that the paper was a bit tough to follow due to the page constraints (much of the content was in the supplemental material), and it could use some proof-reading to fix typos and improve the overall flow.

Despite this, the authors do present novel contributions.

Submitted by Assigned_Reviewer_4

This paper introduces two polytime algorithms for minimum

weight perfect matching using a sequence of linear programs or

runs of max-product BP (the factor graphs for which are

inspired by connections to the LPs). To obtain the polytime

algorithm using BP, the paper makes use of a result from

recent prior work (Park and Shin, 2015): under specific

conditions, for certain pairs of a factor graph and an LP,

running max-product BP on the factor graph converges to the

solution of the LP. The paper proves that the iterative

algorithm uses at most O(n^2) iterations where n is the number

of vertices in the graph.

The authors speculate that their

work may spur further study of BP for MAP on more general

graphical models. Though the paper does not explicitly state

any ideas in this direction, it seems like a furtile area. As

well, the new LP algorithm provides an interesting contrast

with Edmonds' Blossom algorithm, which (in its original form)

required exponentially many constraints. Overall, it's clearly

written, original work that, while not the first polytime

algorithm for min-weight perfect matching (they reserve that

title for (Chandrasekaren et al., 2012)), is likely the

simplest.

It seems the paper would benefit from a journal-style

presentation: one which is more leisurely and example

filled. Given space constraints the paper did a good job of

delegating proofs of certain results to supplementary

material. However, the main paper would benefit greatly if the

examples from Appendix F could be included in a figure, and

repeatedly referenced through the description of the

algorithm.

On the one hand, this paper can be viewed as a nice

application of Park and Shin (2015)'s BP/LP result. On the

other, it stands as a unique example (to the best of my

knowledge as well as claimed by the authors) of solving an ILP

by repeated calls to BP.

This paper does not lack substance, but it would have been

nice to see the algorithm put to use on some real matching

problems. As the linear programming community knows well,

polynomial time algorithms are sometimes less practical than

their exponential time counterparts in the real world.

Misc:

- line 155: Linear Programming --> Linear Program

- line 190: this note deserves a one line explanation

- line 196: initially confusing where T came from

- line 258: if the proof of Theorem 3 will be left in the

appendix, it would help to give some intuition for why the

half-integral solution is obtained.

- line 032 (appendix A): marginal beliefs --> max-marginals

- A more detailed contrast with Chandrasekaren et al. (2012)

seems appropriate

Summary: This work builds on recent work's connections

between max-product BP and linear programming to introduce two

polytime algorithms for minimum weight perfect matching using a

sequence of linear programs or runs of max-product BP (the

factor graphs for which are inspired by connections to the

LPs). Overall, it's clearly written, original work that, while

not the first polytime algorithm for min-weight perfect

matching, is likely the simplest.

Author Feedback
Author rebuttal: We very much appreciate valuable comments, efforts and time spent by the reviewers to evaluate the paper. In what follows, we provide, first, a summary, and then detailed response to each reviewer.

Summary:

1) We have presented the simplest polynomial-time algorithm for solving MWM problem known so far.
-- Our algorithm is simpler than Edmond's algorithms, e.g., Blossom-V, and significantly simpler than the LP-based cutting-plane algorithm developed in [22].

2) Our algorithm is also the first rigorous algorithm solving an Integer Programming (IP) using a sequence of BPs.
-- All prior works on BP are heuristic-based or focused on solving related LP relaxation with no integrality gap (thus only one BP, and not a sequence, is required.)

3) We have also suggested a transparent interpretation of the Edmond's MWM algorithm in terms of a sequence of LPs.
-- Such an interpretation was open for a few decades.

We appreciate very much Reviewer_1/2/3/4 mentioning explicitly "novelty/originality" of our approach and Reviewer_1/6 acknowledging "importance" of our results. Notice that the Edmond's algorithm for MWM was the first complex (going beyond totally unimodular case) polynomial-time algorithm in the computer science literature. The algorithm has motivated the P vs. NP considerations and it has also influenced other combinatorial optimization algorithms (e.g., primal-dual methods). We believe that our novel approach has further potentials to solve a much broader class of IPs using BP, which is of interest to machine learning, in general, and specifically graphical model communities. We also anticipate that our results will be of importance for large-scale machine learning, especially in the context of distributed and parallel implementations.

Response to Reviewer_1:

We are working on computational implementation of our algorithms (both BP and LP) and validating/comparing the algorithms with the Blossom-V, which is the most efficient, modern implementation of the Edmond' algorithm (by Kolmogorov). We have already validated our algorithms over millions of random instances, and found that typically two or three iterations suffices to terminate (while in theory O(|V|^2) iterations may be required in the worst-case). The detailed running time comparison of our algorithms with Blossom V is still incomplete - it is work in progress dependent on many technical details. For example, we have discovered that one can boost up convergence of BP with a targeted initializations and/or damping. This suggests, in particular, using the last message of BP from preceding step to initialize the current state. We have also started to work on developing parallel implementation of the Blossom BP, e.g. testing options provided by GraphLab, GraphChi and OpenMP software. Given the factors mentioned above and also taking into account NIPS page limit constraint, we have decided to publish detailed experimental analysis of our algorithm and performance comparison with Blossom V in a few months when a comprehensive analysis is completed.

Response to Reviewer_2:

Our main theorem implies that the overall complexities of Blossom-BP and Blossom-LP are O(|V|^2) * T_LP and O(|V|^2) * T_BP respectively, where T_LP and T_BP are running times of a single LP and BP algorithms. Obviously, T_LP is polynomial. In fact, one can also prove that T_BP is polynomial as well. In our current draft, we use the result of [16], which does not analyze the number of iterations for BP convergence but instead provides a generic criteria to guarantee convergence of BP. To guarantee that T_BP is polynomial, one can use techniques from [12] or, complementarily, adopt the proof strategy of [16] to analyze convergence time of the algorithm. These proofs are relatively straightforward, however, we have decided not to follow the path in the manuscript as we expect that the worst-case complexity of Blossom-BP cannot beat the best complexity bound known in the literature. Here, we note that Blossom-V also does not beat the best bound, even though it is known as empirically fastest, known algorithm. The main appeal of Blossom-BP and Blossom-LP is in their simplicity, practicality and parallelization potential.

- We do plan to add a number of references summarizing and commenting on related heuristic algorithms and contrasting these with our exact algorithm.
- To the best of our knowledge, there is no work studying LP (5), i.e., a hybrid matching. The half-integrality proof is not too hard, but we have decided to include it for completeness.

Response to Reviewer_3/4/5/6:

- Wrt numerical experiments -- please see our response to Reviewer_1.
- We will correct typos pointed out by the reviewers. We will add a figure illustrating performance of our novel algorithms on examples.

Response to Meta_Reviewer_1:

- Wrt the complexities of our algorithms -- please see our response to Reviewer_2.
- T_LP and T_BP depend on the number of edges.